# Crack Growth Behavior of Additively Manufactured 316L Steel—Influence of Build Orientation and Heat Treatment

**DOI:** 10.3390/ma13153259

**Published:** 2020-07-22

**Authors:** Janusz Kluczyński, Lucjan Śnieżek, Krzysztof Grzelak, Janusz Torzewski, Ireneusz Szachogłuchowicz, Marcin Wachowski, Jakub Łuszczek

**Affiliations:** Faculty of Mechanical Engineering, Institute of Robots & Machine Design, Military University of Technology, 2 Gen. S. Kaliskiego St., 00-908 Warsaw, Poland; lucjan.sniezek@wat.edu.pl (L.Ś.); krzysztof.grzelak@wat.edu.pl (K.G.); janusz.torzewski@wat.edu.pl (J.T.); ireneusz.szachogluchowicz@wat.edu.pl (I.S.); marcin.wachowski@wat.edu.pl (M.W.); jakub.luszczek@wat.edu.pl (J.Ł.)

**Keywords:** additive manufacturing, 316L steel, fatigue cracking, selective laser melting

## Abstract

The effects of build orientation and heat treatment on the crack growth behavior of 316L stainless steel (SS) fabricated via a selective laser melting additive manufacturing process were investigated. Available research results on additively manufactured metallic parts still require a substantial expansion. The most important issue connected with the metal properties after additive manufacturing are the high anisotropy properties, especially from the fatigue point of view. The study examined the crack growth behavior of additively manufactured 316L in comparison to a conventionally made reference material. Both groups of samples were obtained using precipitation heat treatment. Different build orientations in the additively manufactured samples and rolling direction in the reference samples were taken into account as well. Precipitation heat treatment of additively manufactured parts allowed one to achieve microstructure and tensile properties to similar to those of conventionally made pieces. The heat treatment positively affected the fatigue properties. Additionally, precipitation heat treatment of additively manufactured elements significantly affected the reduction of fatigue cracking velocity and changed the fatigue cracking mechanism.

## 1. Introduction

Currently, additive manufacturing (AM) technologies are receiving a lot of attention due to their various applications such as lightweight components or individualized and functionalized parts. The rapid growth of available research results is mainly focused on material analysis, structural tests, tensile and dynamic testing [1,2,3]. The specific layered structure of an additively manufactured element affects the high anisotropy of the material [4,5] which plays a significant role from a fatigue performance point of view. Additionally, some research papers discuss powders usage [6] or additional energy sources [7] in various applications to improve the fatigue life.

One of the most interesting technologies is selective laser melting, which is a well-known laser powder bed fusion technology (LPBF) that allows one to obtain very dense parts, characterized by very good mechanical properties [8]. Material fusion with a high-temperature gradient (between the melting pool and process chamber atmosphere) and layer-wise building affect the material anisotropy, which plays a significant role during fatigue loading of select laser melted parts. There is a volume of available research results connected to fatigue analysis of elements manufactured using selective laser melting [9,10,11]. A lot of research facilities are focused on titanium alloys and nickel alloys which are of great interest to the aerospace and military industry. In this study, 316L stainless steel was taken into account. This material is characterized by improved weldability and good corrosion-resistance properties caused by its lower amount of carbon (in comparison to 316 stainless steel).

These properties make 316L steel a very good material to use in various industries—marine, aircraft, food, medical, pharmaceutical, chemical, or automotive [12,13,14,15].

The great interest in AM technologies has led to the appearance of many studies about selectively laser melted 316L steel parts. Some literature concerning the fatigue properties of this material, including crack growth properties, is also available, but the amount of that type of research results is small [11,16,17,18,19,20].

Fergani et al. [16] studied the effect of build orientation on the crack growth by performing fatigue crack growth tests on compact tension samples built at 0° and 45° orientations relative to the build direction with additional heat treatment above the recrystallization temperature followed by quenching. The obtained results allowed the authors to state that proper heat treatment improved the crack growth resistance but at the same time it did not remove the material anisotropy. Riemer et al. in their research [18] stated that high ductility of selective laser melted 316L stainless steel shows fatigue properties similar to those of conventionally processed material in its as-built condition, and the material may be used in applications without post-processing operations. Suryawanshi et al. [21] examined the crack growth behavior of select laser melted 316L steel samples in terms of the meso- and microstructural features. During their research, no significant anisotropy in the mechanical properties was found, which was motivated by strong intra-layer bonding and the absence of a dominant texture.

The abovementioned literature thus contains conflicting information about the crack growth behavior of selective laser melted 316L steel. Additionally, none of the available research results contain references to conventionally made material. These issues were our motivation to initiate our own research to address this lack of knowledge about the build orientation and heat treatment of selective laser melted 316L steel compared to conventionally made 316L steel (cold-rolled sheets). To assure reliable results, conventionally made material was also examined under conditions of different orientation of the rolling direction with additional heat treatment of half of the samples.

## 2. Materials and Methods

### 2.1. Material

The powder (purchased from LPW Technology, Ltd., Cheshire, UK) used for the production of all samples was gas atomized steel 316L (1.4404) in an argon atmosphere. Based on the own scanning electron microscopy (SEM) results (shown in Figure 1), the powder particles were spherical, with diameters of 15–63 µm.

The density of the material was 7.92 g/cm^3^ and its flowability was 14.6 s/50 g. The cumulated mass value of the powder particles size distribution has the following values: D10 = 18.22 μm, D50 = 30.50 μm, D90 = 55.87 μm. The nominal material chemical composition is shown in Table 1.

As a reference material conventionally made 316L steel (cold-rolled sheet) was used.

### 2.2. Description of the Manufacturing Process

To produce the test parts a SLM 125HL system (SLM Solutions, Lubeck, Germany) was used. The production process is fully-automated and base on 3D-CAD data. The 3D model is saved as a stereolithography tessellated file (.stl). A job file was prepared using the Magics software (version 19, Materialise, Leuven, Belgium) with an additional metal build processor (MBP) module dedicated to the SLM technology. After file preparation, the process had been run in an argon atmosphere with the oxygen level in the machine’s build chamber below 0.1%. For the AM process the following parameters were used: Laser power: 190 W,Exposure velocity: 900 mm/s,Hatching distance (distance between exposure lines): 0.12 mm,Layer thickness: 0.03 mm

The mentioned parameters generated an energy density of 58.64 J/mm^3^–based on the following Equation:(1)$$\rho_{e}=\frac{L_{p}}{e_{v}\times h_{d}\times l_{t}}$$
where: L_P_–laser power [W], e_v_–exposure velocity [mm/s], h_d_–hatching distance [mm], and l_t_–layer thickness [mm].

For fatigue cracking velocity determination, compact Compact Tension (CT) samples were developed based on the ASTM E647 standard. The development of model elements was also based on the study [22], in which this type of research was carried out on elements additively manufactured using selective laser melting technology. Samples with the dimensions shown in Figure 2 were made in such a way to allow fatigue testing in a perpendicular, parallel or angled direction (at an angle of 45°) according to the plane parallel to the substrate plate surface.

The aforementioned build orientations with indicated loading direction are shown in Figure 3. To obtain designed test conditions and reliable results, samples were manufactured in a single process with the orientation shown in Figure 4.

### 2.3. Description of the Testing Methodology

An Instron 8802RS testing machine (Instron, Norwood, MA, USA) with dedicated *da/dN* Fatigue Crack Propagation (*da/dN* FCP) software was used for the research. Standardized compact (CT) samples with additional fatigue pre-crack were subjected to cyclically variable loading with a constant force amplitude. The load range was 2500 N for an average stress level characterized by an asymmetry load *R* = 0.1 with a sinusoidal shape of the load cycle at load changes frequency *f* = 10 Hz. During the test, the number of *N* cycles, crack opening displacement (COD), and applied force was recorded. A minimum of three samples was tested in each test series. The crack length was on-line-measured based on crack cleavage measurements using a COD extensometer and a sample compliance changing method. Crack length measurements were additionally tracked using an optical microscope with an x10 magnification equipped with a movable table equipped with micrometer screws. The range of stress intensity coefficient (∆*K*) was calculated using *da/dN* FCP software based on Equation (2) proposed by the ASTM E647 standard:(2)$$∆K=\frac{∆P\left(2+\alpha \right)}{B\sqrt{W}{\left(1-\alpha \right)}^{\frac{3}{2}}}\left(0.886+4.64\alpha -13.32\alpha^{2}+14.72\alpha^{3}-5.6\alpha^{4}\right)$$
where: α is the normalized crack length (α = *a/W*), B is the sample thickness (mm), *W* is the sample width(mm) and ∆*P* is the force amplitude (N).

The surface structures of the sample fractures after the tensile tests were observed using a Jeol JSM-6610 SEM scanning electron microscope (SEM, JEOL Ltd., Tokyo, Japan). The microstructural investigation has been performed using an Olympus LEXT 4100 confocal microscope (Olympus Corporation, Tokyo, Japan).

The heat treatment used was a solution annealing performed under a temperature of 1060 °C for 6 h using a Nabertherm P300 annealing furnace (Nabertherm GmbH, Lilienthal, Germany). To reduce the formation of high-dimensional grains, water cooling of the samples was applied directly after annealing.

## 3. Results and Discussion

### 3.1. Microstructure Analysis

The first-stage analysis related to a microstructural investigation of all tested samples (Table 2). Selectively laser melted parts are characterized by a typical, layered structure with visible melting pools and columnar grains [17]. That kind of microstructure is caused by the rapid solidification, which significantly increases the tensile strength of additively manufactured parts. Heat treatment of selective laser melted parts caused the disappearance of a layered structure of the material and make it similar to conventionally made material.

Analysis of the conventionally made material showed a reduction of the visible rolling direction and reduced the length of visible grains. A reduction of twins in the structure of the material after applying the heat treatment is also visible. An obtained microstructure of selective laser melted parts shows a mesoscopic structure, with hatch overlapped regions in the form of half-cylindrical contours, which were affected by the AM process. Selective laser melted samples reveal a fine-grained structure with grains elongated along the direction of thermocapillary convection in the melted pool and heat dissipation. In contrast to the selective laser melted 316L steel, a CM alloy with similar composition would contain FCC austenite grains of 30–80 µm size [11].

### 3.2. Static Tensile Testing Results

Axial tensile strength tests of the additively manufactured samples made of 316L steel were carried out according to ASTM E466-96 with the use of an Instron 8802 hydraulic pulsator (Instron, Norwood, MA, USA). Measurements of elongation under axial stretching were obtained using an Instron 2630-112 extensometer with a measuring base of 25 mm. All samples subjected to axial tension had the same geometry.

To increase the accuracy of inferences regarding the effect of heat treatment on the tested materials, tensile tests of samples were performed. The results are shown in Table 3. The standard deviation for 0.2% offset yield strength (YS) is given at ± 6 MPa, for ultimate tensile strength (UTS) ± 6 MPa, elongation at failure (ε_f_) ± 1.81 % and Young modulus (E) ± 2.1 GPa.

Selective laser melted samples are characterized by about 40% higher YS, about 20% higher UTS (ultimate tensile strength), and about 30% lower elongation at failure in comparison to conventionally manufactured samples.

The mechanical properties listed in Table 3 indicate a significant benefit based on increased YS, which could be attributed to the marked refinement in the microstructure, which is a result of the high cooling rates (~103–108 K/s) achieved during the selective laser melting. However, a higher level of UTS in comparison with CM material is significantly smaller, which indicates that selective laser melted parts does not display the same level of work hardening as the cold-rolled CM parts made of the same material. Analyzing the tensile stress-strain plots available from our own previous research [3] allows us to indicate that the selective laser melted part’s stress-strain response is nearly elastic-perfectly plastic, with very small strain hardening. Moreover, geometrically complex shapes that can be obtained using selective laser melting would not be subject to further forming using rolling or forging on the produced components. For that reason, the reduction in ductility need not necessarily be viewed as a serious detriment when it comes to the application potential of the selective laser melted 316L steel.

Solution annealing of selective laser melted samples decreased the YS and UTS strength properties with a simultaneous increase in ε_f_. This phenomenon could be related to a significant modification of the material structure, where there is a reduction of fine-grain after selective laser melting connected with very high cooling rates and substantial strain hardening–stress-induced austenite to martensite transformation (SIMT) [11].

Additional heat treatment of conventionally manufactured 316L steel caused a lowering the YS by about 40% and an increase the total strain of 10%, which could be also connected with the removal of the structures obtained during cold rolling.

### 3.3. Crack Growth Behavior

A significant influence of additional heat treatment on fatigue crack growth was registered in the case of additively manufactured samples only. The results for crack length in a function of the number of cycles are given in Figure 5a–c.

Growth in the total number of cycles in selective laser melted and solution annealed samples in comparison with the same parts without any heat treatment could be observed. The same phenomenon was observed in the research results published by Fergani et al. [16]. The smallest influence of heat treatment was registered in vertically oriented samples–where the loading direction was parallel to the layer deposition direction (Figure 5b). The biggest growth in fatigue life was registered for 45°-degrees angled samples (Figure 5c). In those samples, crack length growth was slightly increased during about 85% of the total fatigue life of tested parts. In the last 15% of fatigue life, the crack length progressed significantly. This phenomenon was proven by the preparation of charts including crack growth rates and the corresponding stress intensity ranges (Table 3). It could be observed a significant tendency of increased cracking velocity in all selective laser melted and heat-treated samples, especially in 45°-angled ones. That characteristic cracking behavior on in 45°-angled samples could be related to a natural barrier made of layered and angled material structure. At the beginning of fatigue loading in the aforementioned samples, there was no visible significant crack propagation. At the same time, a lot of microcracks appeared in the material structure, which is visible as characteristic white lines in attached SEM images in Table 4. After a specific number of cycles 45°-angled samples, the cracking process became very dynamic through all imperfections like pores and non-melted powder grains.

During the selective laser melted steel horizontally oriented samples fracture observations, it was possible to discover a higher number of pores and grains along the whole crack area. This was probably caused by structural heterogeneity connected with local stress damming, which promoted microcracking initiation. It is possible to observe that cracking went through several layers which finally formed an irregular, multiplanar crack.

During the cyclical loading, some microcracks connections caused plastic deformations and material decohesion. Solution heat treatment in the case of selective laser melted samples caused the reduction of microcracks and lowered the number of cracks in different layers. After solution annealing, the cracking velocity increased slightly, which could be connected with porosity growth after heat treatment.

In the case of selective laser melted, vertically oriented samples, fatigue crack propagation frontlines went along the build direction vectors. Crack lines are accumulated near defects (pores and non-melted grains). Based on SEM images it is possible to observe an increase in microcracks after heat treatment, which caused more dynamic cracking behavior after a specific number of cycles.

Comparing all selective laser melted samples results with reference samples made of conventionally manufactured material a high homogeneity of rolled 316L steel sheets is visible. Additional heat treatment caused an increase in material plasticity without a significant change in fatigue cracking velocity. Basing on classical Paris equation:*da/dN* = *C*(∆*K*) ^*m*^(3)
where, *C*–coefficient, and *m*-coefficient are material coefficients determined by material properties, loading type, and sample’s geometry. The mentioned coefficients were determined for each sample and shown in Table 5.

The determined m–coefficientd are helpful to identify a tendency for cracking velocity in each sample. After selective laser melting, the smallest cracking velocity value was registered in vertically oriented samples. The same phenomenon was registered for horizontally oriented reference samples. The opposite situation is visible after the heat treatment of selective laser melted samples, where the m–coefficient increased almost two-fold. This phenomenon could be related to porosity growth after solution annealing which was proven by additional microscopic observations and can be observed in the SEM micrographs shown in Table 4. Loading across layers connections caused favorable conditions for cracking initiation between layers. After the heat treatment of horizontally oriented reference material, the m–coefficient decreased. This corresponds with values in Table 2 where it is visible that after solution annealing, the plasticity of the reference material increased significantly. Horizontally-oriented SLM–processed samples had very similar cracking characteristics compared to the reference material.

The same material properties phenomenon after heat treatment was registered for all horizontally oriented samples: selective laser melted samples were characterized by increased m – coefficients, wheras in reference material, this value decreased.

Vertically oriented samples were characterized by the lowest cracking velocity but also a low fatigue life of the tested parts. Fatigue crack propagation frontlines went along build direction vectors. Crack lines are accumulated near defects (pores and non-melted grains). Based on the SEM images it is possible to observe an increase in microcracks after heat treatment, which caused more dynamic cracking behavior after a specific number of cycles.

45°-angled samples were characterized by the highest value of cracking velocity in all tested combinations, which was caused by increasing porosity between the deposited layers of the material, but in the case of selective laser melted samples, the highest fatigue life. This phenomenon was determined by the presence of a natural barrier for crack propagation in a form of angled material layers. During cyclic loading of samples, a lot of microcracks appeared which caused dynamic cracking after the mentioned microcracks merge. The described phenomenon can be observed in the SEM micrograph in Figure 6, showing the presence of fatigue striations that propagate till the cracking plane in the material changes.

In Figure 6 are shown fatigue striations that propagate in different cracking planes until its merge and cause brittle-like cracking. Such a phenomenon is conducive to increased cracking velocity and changes the cracking character of the material. Additionally, the presence of non-melted grains and pores in the material volume introduces structural heterogeneity which causes a negative, local stress concentration.

The mechanism of cracking before heat treatment is affected by the layered structure of the material and direction of layer deposition because of quite different material properties around the layer fusion area. This phenomenon was visible in our previous own research [3] where the structure of the selective laser melted parts was examined in the field of sclerometric hardness in different planes (regarding layer deposition direction). As it could be seen in results shown in [3] there is a slight increase in visible sclerometric hardness which could affect the stress concentration near layer borders. After additional heat treatment, the layered structure disappeared but still, some imperfections (non-melted grains and porosity) were visible. Additionally, during the heat treatment of austenitic steels, there is a significant issue connected with the brittle-like sigma phase occurrence. Proper heat treatment of this material could significantly reduce the mentioned phase generation. A significant amount of microcracks visible after heat treatment of conventionally and additively manufactured 316L steel parts could be caused by the presence of some amount of mentioned sigma phase or carbide-nitrides M_2_X [23,24].

## 4. Conclusions

The research results presented in this paper are, in the authors’ opinion, helpful to understand additively manufactured 316L steel behavior in comparison with conventionally manufactured material. Additional data connected with material properties after heat treatment expanded the scope of the research and allowed for an extension of the formulated conclusions. Baed on the recorded and described research results, the following conclusions may be formulated:(1)Microstructural investigation enabled us to expose that after removal of a typical, layered structure of the material after heat treatment of additive manufactured parts a different shaped-grains were obtained. This caused a fatigue life increase but also generated a lot of microcracks which caused a rapid cracking phenomenon. A similar phenomenon was noticed after conventionally made material heat treatment, where the shape and size of elongated grains changed after material rolling and this affected the appearance of microcracks in the material volume during fatigue loading.(2)Proper orientation of parts in additive manufacturing allows one to achieve a fatigue life and cracking growth behavior similar to that of conventionally made material.(3)Solution heat treatment of additively manufactured 316L steel increases the fatigue life of produced parts, but completely changes the cracking growth behavior. That kind of material has no significantly visible cracking symptoms. In the material volume, many microcracks appear during cyclic loading which can merge, thus changing the cracking character.(4)The presence of porosity and non-melted grains introduces a structural heterogeneity that causes a negative, local stress concentration. That phenomenon directly influences the change in characteristic cracking from plastic fatigue cracking with visible fatigue striations to dynamic, brittle-like cracking.(5)Heat treatment of additively manufactured completely changes the fatigue crack behavior and total fatigue life of the processed part. The influence of solution annealing on AM materials is opposite to that of the same heat treatment of the same material, but manufactured conventionally.

## Figures and Tables

**Figure 1 materials-13-03259-f001:**
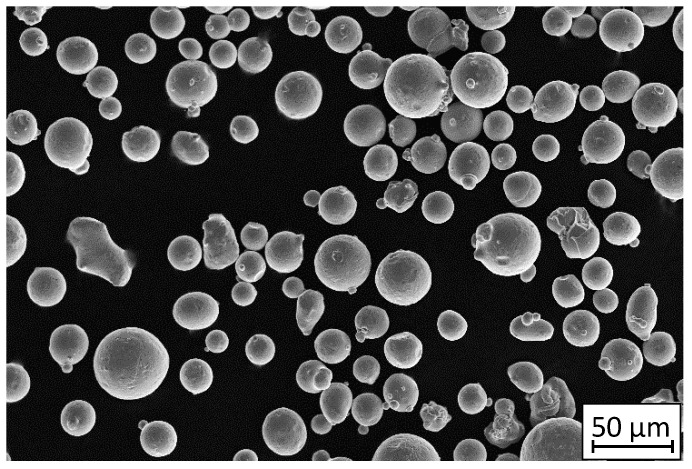
SEM image of 316L powder grains captured on a 50 µm scale.

**Figure 2 materials-13-03259-f002:**
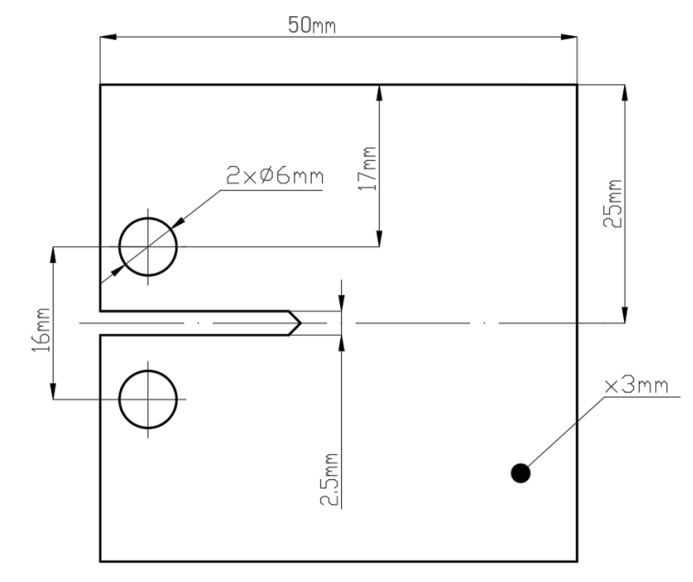
CT sample for fatigue cracking velocity tests.

**Figure 3 materials-13-03259-f003:**
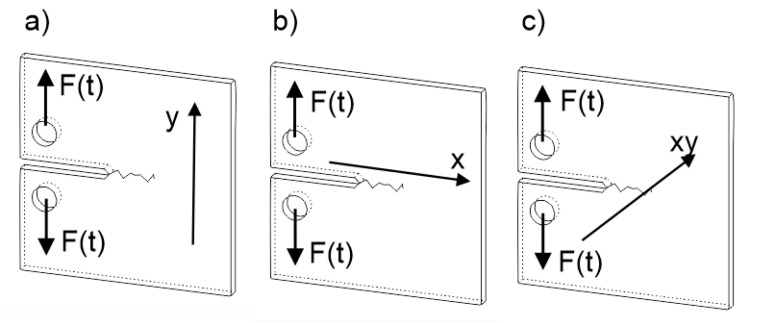
Fatigue load direction F(t) with the layer deposition direction (x): (**a**) perpendicularly to the layer deposition direction (H), (**b**) parallelly to the layer deposition direction (V), (**c**) 45° angled to the layer deposition direction (A), F(t)–loading direction.

**Figure 4 materials-13-03259-f004:**
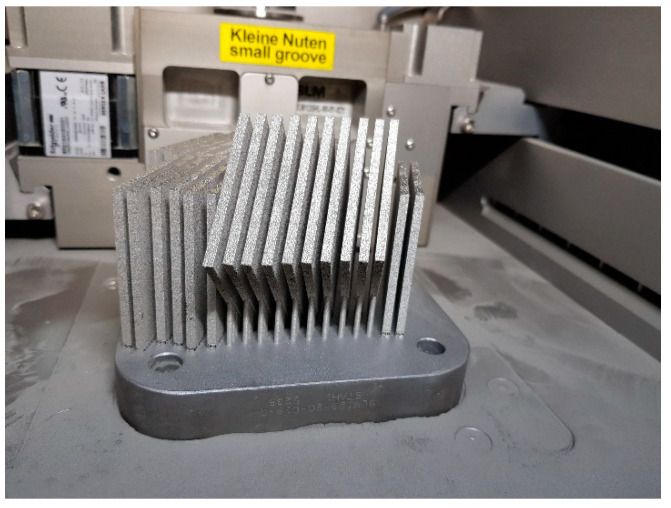
CT samples on the SLM 125HLs’ device substrate plate.

**Figure 5 materials-13-03259-f005:**
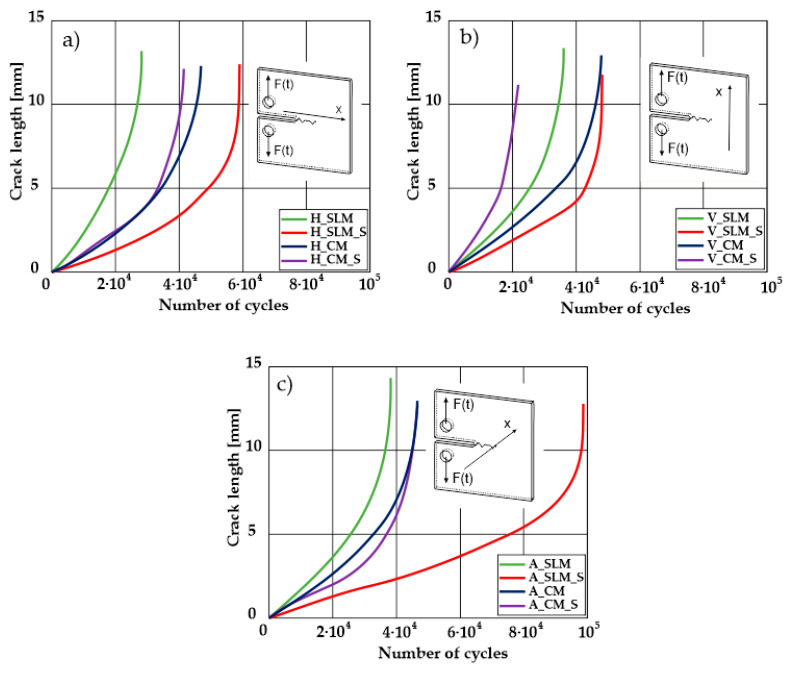
Fatigue crack length as a function of the number of cycles to failure for all tested material configurations in three different build/rolling orientations. (**a**) Horizontally oriented samples; (**b**) Vertically oriented samples; (**c**) 45° angled samples.

**Figure 6 materials-13-03259-f006:**
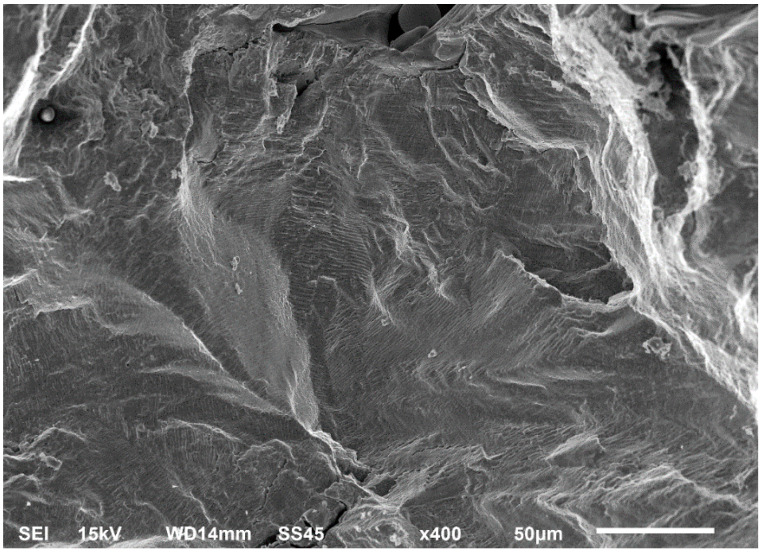
Additively manufactured, 45° angled sample (A_SLM) with an example of fatigue striations.

**Table 1 materials-13-03259-t001:** 316L steel nominal chemical composition (weight [%]).

C	Mn	Si	P	S	N	Cr	Mo	Ni
max. 0.03	max. 2.00	max. 0.75	max. 0.04	max. 0.03	max. 0.10	16.00–18.00	2.00–3.00	10.00–14.00

**Table 2 materials-13-03259-t002:** The microstructure of all types of tested samples.

Condition	Microstructure
Selective laser melted (SLM)	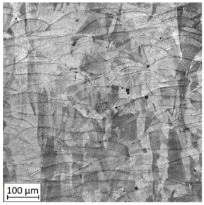
Selective laser melted and solution annealed (SLM_S)	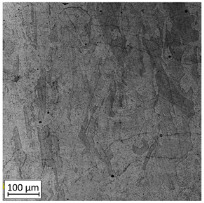
Conventionally manufactured (CM)	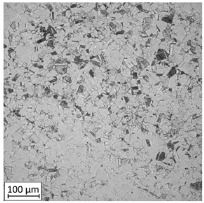
Conventionally manufactured and solution annealed (CM_S)	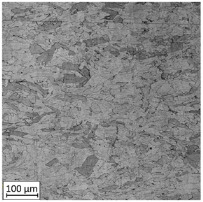

**Table 3 materials-13-03259-t003:** Mechanical properties of tested samples manufactured parallelly to the layer deposition direction.

Condition	0.2% YS (MPa)	UTS (MPa)	ε_f_ (%)	E (GPa)
Selective laser melted (SLM)	554	666	44.91	169.9
Selective laser melted and solution annealed (SLM_S)	371	632	47.35	170.6
Conventionally manufactured (CM)	346	564	64.11	204.5
Conventionally manufactured and solution annealed (CM_S)	215	567	69.95	205.3

**Table 4 materials-13-03259-t004:** Crack growth curves and fatigue fracture surfaces of examined samples.

Crack Growth Curves			Fracture Surfaces
Horizontal direction of layers deposition/sheet rolling 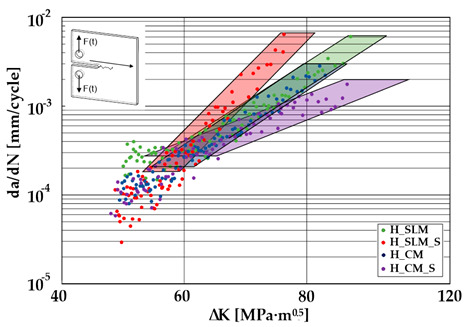	Material type	H_SLM	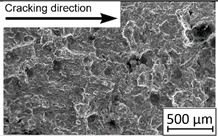
		H_SLM_S	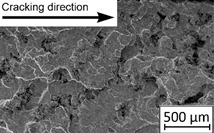
		H_CM	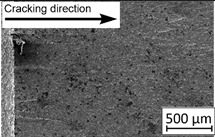
		H_CM _S	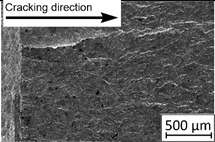
Vertical direction of layers deposition/sheet rolling 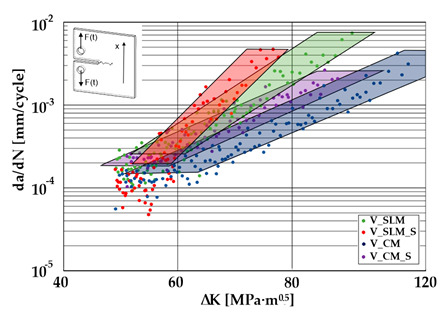	Material type	V_SLM	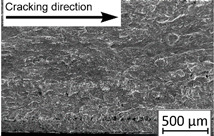
		V_SLM_S	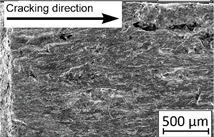
		V_CM	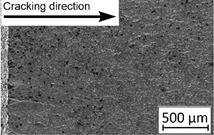
		V_CM _S	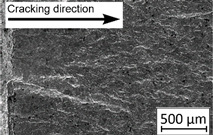
45°-angled direction of layers deposition/sheet rolling 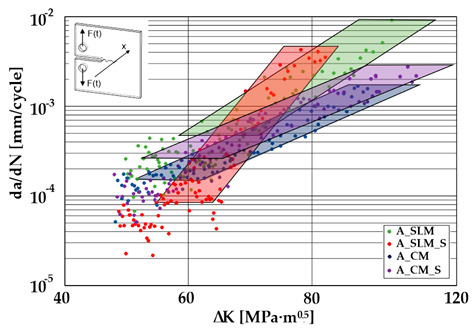	Material type	A_SLM	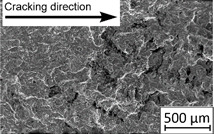
		A_SLM_S	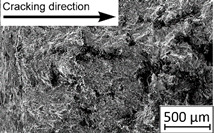
		A_CM	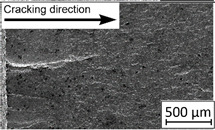
		A_CM _S	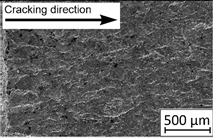

**Table 5 materials-13-03259-t005:** Paris law parameters for 316L SLM processed and conventionally manufactured

Condition	Coefficient Type	Horizontally Oriented	Vertically Oriented	45° Angled
Selective laser melted (SLM)	C-coefficient	7 × 10^−15^	2 × 10^−14^	5 × 10^−15^
	m-coefficient	6.00	5.86	6.06
Selective laser melted processed and solution annealed (SLM_S)	C-coefficient	3 × 10^−20^	10^−18^	5 × 10^−28^
	m-coefficient	9.16	8.27	13.14
Conventionally manufactured (CM)	C-coefficient	5 × 10^−15^	5 × 10^−11^	6 × 10^−12^
	m-coefficient	6.08	3.72	4 × 24
Conventionally manufactured and solution annealed (CM_S)	C-coefficient	8 × 10^−11^	2 × 10^−11^	5 × 10^−10^
	m-coefficient	3.74	4.09	3.30

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
