# Peer review of "Crack Growth Behavior of Additively Manufactured 316L Steel—Influence of Build Orientation and Heat Treatment"

_materials, 2020, doi:10.3390/ma13153259_

Round 1

Reviewer 1 Report

The topic is nice but it requires significant improvements.

The details of the SLM process and heat treatment are missing, so they must be added.

The correlation between the mcirostructure and mechanical properties are not highlighted.

Table 2 seems containing data from literature. Is it fine? If yes, please add the references

Author Response

Dear Reviewer,

In the beginning, we would like to thank you for reading our paper, its deep analysis, and your correction suggestions. We have attached corrections based on your suggestions. Proper corrections were yellow-highlighted and improved text lines were put in each comment. 

“The topic is nice but it requires significant improvements.”

Ad.1. As a topic of our manuscript, we understood the full volume of our article. Basing on your advice we provided required corrections which we have yellow-highlighted and mentioned text lines where it has been put.

“The details of the SLM process and heat treatment are missing, so they must be added.”

Ad. 2. Regarding SLM technology we put additional description (lines 83-89) which were yellow-highlighted. In connection with heat treatment, we provided, in our opinion, sufficient data about used heat treatment - please have a look at lines 131-134.

“The correlation between the microstructure and mechanical properties are not highlighted.”

Ad. 3. We made additional microstructural investigation – please have a look at chapter 3.1. We described there a microstructure of all tested samples - lines 137-146, and put additional conclusion: lines 270-275.

“Table 2 seems containing data from literature. Is it fine? If yes, please add the references”

Ad.4. All provided data is base on our preliminary tensile tests – for that reason, we also provided information about standard deviation before the mentioned table. We improved this table by attaching the column with Young’s modulus.

Sincerely, 

Authors

Reviewer 2 Report

The authors submit a very interesting study on fatique performance of 316L Steel AM-manufactured parts in comparison with conventionally manufactured parts. They focus on investigations of the cracking behavior in dependence of AM-direction and thermal treatment.

The background and methods are adequately described, the language is clear and understandable and the results are quite intelligible.

However, the study is very applied. A discussion of possible mechanisms is missing. The authors could add a paragraph on why cracking depends on the direction of AM and thermal treatment in view of relevant. For example it is known for carbon steel, that hardening of the material renders it more brittle. Is this the case for AM-manufactured 316L-part too? How do they compare to CM parts in light of this basic principle?

Author Response

Dear Reviewer,

In the beginning, we would like to thank you for reading our paper, its deep analysis, and your correction suggestions. We have attached a corrected based on your suggestions. Proper corrections were yellow-highlighted and improved text lines were put in each comment. 

“(…) A discussion of possible mechanisms is missing. The authors could add a paragraph on why cracking depends on the direction of AM and thermal treatment in view of relevant. For example, it is known for carbon steel, that hardening of the material renders it more brittle. Is this the case for AM-manufactured 316L-part too? How do they compare to CM parts in light of this basic principle?”

We put additional discussion connected with layer deposition direction, it was yellow-highlighted at lines 250-262, based on the following literature:

Lo, K.H.; Shek, C.H.; Lai, J.K.L. Recent developments in stainless steels. Mater. Sci. Eng. R Reports 2009, 65, 39–104.

Vach, M.; Kuníková, T.; Dománková, M.; Ševc, P.; Čaplovič, Ľ.; Gogola, P.; Janovec, J. Evolution of secondary phases in austenitic stainless steels during long-term exposures at 600, 650 and 800 °C. Mater. Charact. 2008, 59, 1792–1798.

We would like to thank you for your suggestion about a possible phenomenon as it is in carbon steel. Unfortunately, in our material, there is a different issue, especially connected with porosity, but also with non-melted grains. Additionally, in austenitic steel for example 316L steel there is a possibility of sigma phase generation – it is a very hard and brittle phase that negatively affects material properties.  We have described it in the attached part of our manuscript.

Sincerely,

Authors.

Reviewer 3 Report

This work reports a comparison of 316L steel prepared by conventional and by additive manufacturing (AM) methods, and on the effects of solution melting on crack propagation for vertical, horizontal and diagonal stress loadings. Results are very interesting and indicate the effects of pores present in materials where poor melting was achieved during manufacturing. Authors further obtain a great number of data points for crack propagation, Paris equation parameters and including the ultimate mechanical strength, yield strength and elongation at failure. This work is of interest to materials scientists and engineers, including mechanical, civil, aerospace, etc., working on steel, alloys, manufacturing methods, and applications. I recommend this work for publication with minor recommendations. Once comments are addressed, no further revisions are needed. Comments:

  1. Language requires minor improvements.
  2. Do authors have the elastic modulus of each sample to include in Table 2?
  3. Include sample used for image in Figure 6 caption.

Author Response

# Reviewer 3

Dear Reviewer,

In the beginning, we would like to thank you for reading our paper, its deep analysis, and your correction suggestions. We have attached a corrected based on your suggestions. Proper corrections were yellow-highlighted and improved text lines were put in each comment. 

“Language requires minor improvements.” 

We have read our manuscript and discussed it with our specialist from the foreign languages department – we corrected all the linguistic issues which we were able to found.

“Do authors have the elastic modulus of each sample to include in Table 2?”

Yes. We put an additional column in table 2.

“Include sample used for image in Figure 6 caption”

We put a sample image near the fatigue striations image to follow your advice.

Sincerely,

Authors.

Reviewer 4 Report

Ref_comments to the paper with Manuscript ID: materials-858998 titled as
“Crack growth behavior of additively manufactured 316L steel –influence of build orientation and heat treatment” written by the authors: Janusz Kluczynski, Lucjan Śnieżek, Krzysztof  Grzelak, Janusz  Torzewski, Ireneusz Szachogłuchowicz, Marcin Wachowski, Jakub Łuszczek.

It is known that steel materials are widely used in various fields of technology and human life. Naturally, the operation of steel products under different external conditions leads, among other things, to the formation of cracks, which significantly affects the durability. From this point of view the paper is actual and modern. It is worth noting that the authors have analyzed scientific and technical publications of the last 3 years and clearly understand the place of their research among the works of other teams.

The paper presents the fine data about the chemical composition of the studied steel, its good SEM-image, description of the manufacturing process and the methodology testing, analysis of the mechanical features and fatigue crack length as a function of the number of cycles to failure for all tested material. It should be remarked that the crack growth curves and fatigue fracture surfaces of examined samples are correct and logically analyzed. Moreover, the data about obtained Paris law parameters for 316L SLM processed and conventionally manufactured are extended our knowledge in this direction.

The conclusion is accumulated the basic results obtained by the authors of this paper.

Indeed, the job is interesting and can be recommended to the publication after minor corrections.

The recommendation is the following:

1). I would like to recommend that the authors should correct the abbreviation SLM for the process of the selective laser melting. The fact is that this abbreviation has been used for spatial light modulators (SLM) for a long time. This is accepted and established symbol both in a number of countries and in the world scientific arena. Therefore, it is not quite logical to enter the same symbol for the process considered by the author of this article

2). Please tilt the Latin characters in formulas and leave Greek characters straight.

Author Response

# Reviewer 4

Dear Reviewer,

In the beginning, we would like to thank you for reading our paper, its deep analysis, and your correction suggestions. We have attached a corrected based on your suggestions.

“1). I would like to recommend that the authors should correct the abbreviation SLM for the process of the selective laser melting. The fact is that this abbreviation has been used for spatial light modulators (SLM) for a long time. This is accepted and established symbol both in a number of countries and in the world scientific arena. Therefore, it is not quite logical to enter the same symbol for the process considered by the author of this article”

Following your advice, we have removed the SLM abbreviation and use the full name of the process.

“2). Please tilt the Latin characters in formulas and leave Greek characters straight.”

We made proper corrections. Thank you.

Sincerely,

Authors.

Round 2

Reviewer 1 Report

The reviewer would like to thank the authors for the revisions implemented in the manuscript.

There are two points not well addressed:

  • details of the SLM process are not reported: the sentence reported in yellow is just general, so not necessary. Please, report, the model of the system, the powder characteristics and the main process parameters.
  • describe better the mecchanical properties in tensile from the table
  • the correlation between the obtained microstructures and the relative mechanical properties is not described

Author Response

Dear Reviewer,

Regarding your comments we have made the following corrections (green highlighted):  

  1. “Details of the SLM process are not reported: the sentence reported in yellow is just general, so not necessary. Please, report, the model of the system, the powder characteristics, and the main process parameters.”

Ad.1. SLM system and process description were attached, please look at the lines:

  • 72-77 for powder,
  • 85-101 for SLM system and process parameters.
  1. “Describe better the mecchanical properties in tensile from the table”

Ad.2 We have provided additional description – please have a look at lines 166-170; 178-198

  1. “the correlation between the obtained microstructures and the relative mechanical properties is not described”

Ad.3 We improved this topic please have a look at lines 159-164 and 178-198.